# Quantitative Autofluorescence in Non-Neovascular Age Related Macular Degeneration

**DOI:** 10.3390/biomedicines11020560

**Published:** 2023-02-15

**Authors:** Shruti Chandra, Manjot K. Grewal, Sarega Gurudas, Rajan Sondh, Alan Bird, Glen Jeffery, Victor Chong, Sobha Sivaprasad

**Affiliations:** 1Institute of Ophthalmology, University College London, London EC1V 9EL, UK; 2Moorfields National Institute of Health and Care Research Facility, Moorfields Eye Hospital, London EC1V 2PD, UK; 3Sandwell and West Birmingham Hospitals NHS Trust, Birmingham B71 4HJ, UK

**Keywords:** age-related macular degeneration, quantitative autofluorescence, subretinal drusenoid deposits, hyperreflective foci, drusen, geographic atrophy

## Abstract

Quantitative autofluorescence (qAF8) level is a presumed surrogate marker of lipofuscin content in the retina. We investigated the changes in the qAF8 levels in non-neovascular AMD. In this prospective cohort study, Caucasians aged ≥50 years with varying severity of non-neovascular AMD in at least one eye and Snellen visual acuity ≥6/18 were recruited. The qAF8 levels were analysed in the middle eight segments of the Delori pattern (HEYEX software, Heidelberg, Germany). The AMD categories were graded using both the Beckman classification and multimodal imaging (MMI) to include the presence of subretinal drusenoid deposits (SDD). A total of 353 eyes from 231 participants were analyzed. Compared with the age-matched controls, the qAF8 values decreased in the eyes with AMD (adjusted % difference = −19.7% [95% CI −28.8%, −10.4%]; *p* < 0.001) and across the AMD categories, (adjusted % differences; Early, −13.1% (−24.4%, −1%), *p* = 0.04; intermediate AMD (iAMD), −22.9% (−32.3%, −13.1%), *p* < 0.001; geographic atrophy −25.2% (−38.1%, −10.4%), *p* = 0.002). On MMI, the qAF8 was reduced in the AMD subgroups relative to the controls, (adjusted % differences; Early, −5.8% (−18.9%, 8.3%); *p* = 0.40; iAMD, −26.7% (−36.2%, −15.6%); *p* < 0.001; SDD, −23.7% (−33.6%, −12.2%); *p* < 0.001; atrophy, −26.7% (−39.3%, −11.3%), *p* = 0.001). The qAF8 levels declined early in AMD and were not significantly different between the severity levels of non-neovascular AMD, suggesting the early and sustained loss of function of the retinal pigment epithelium in AMD.

## 1. Introduction

Bisretinoid fluorophores in the phagocytosed photoreceptor outer segments are by-products of the visual cycle that accumulate as lipofuscin within the human retinal pigment epithelium (RPE). Lipofuscin is a major contributor of fundus autofluorescence (FAF). An age-related non-linear increase in the amount of autofluorescence in RPE has been observed, with significant variations in the same age-group [1]. Delori et al. confirmed that lipofuscin fluorescence increases linearly until age 70 in healthy eyes, then declines, probably due to the loss of RPE cells [2]. 

Excessive lipofuscin could adversely affect the essential RPE functions and contribute to the pathogenesis of age-related macular degeneration (AMD) [3,4,5]. The phototoxic and proinflammatory effects of lipofuscin may cause RPE cell death in AMD eyes [6,7]. For example, hyper-autofluorescence, seen around the border of geographic atrophy (GA), may be a sign of excessive lipofuscin that precedes atrophy or a displacement or stacking up of RPE cells with the accumulated lipofuscin [8]. The presence of lipofuscin in drusen also suggests that it may be involved in drusen biogenesis [9]. By contrast, the loss of lipofuscin from RPE cell death is seen as hypo-autofluorescence, as is typically seen in GA. 

Quantitative fundus autofluorescence (qAF8) enables the in vivo quantification of these FAF signals compared with an internal standard. It has been concluded that eyes with increased qAF8 levels may benefit from anti-lipofuscin therapies [5,10,11]. Studies have reported qAF8 levels in AMD eyes and its correlation with factors such as age, sex, ethnicity, smoking status, type and volume of drusen [12,13,14,15]. These reports consistently showed either normal or reduced qAF8 levels in AMD eyes compared with age-matched controls. It is unclear whether the reduced qAF8 levels are associated with the loss of function and integrity of RPE cells or because less lipofuscin is produced due to abnormalities in the visual cycle [16]. As subretinal drusenoid deposits (SDD) are associated with a prolonged rod-intercept time, it may be a surrogate for impaired visual cycle.

Deciphering the relation of qAF8 to the RPE and the visual cycle is important as there are interventional trials for GA aimed to modulate the visual cycle or reduce lipofuscin. Examples include oral LBS-008 (Tinlarebant), a small molecule retinol binding protein 4 (RBP4) antagonist that is being evaluated in Stargardt Disease. Its effect on AMD is unknown. Another RBP4 inhibitor, fenretinide, showed a correlation with a reduction in the GA growth rate, but did not delay the progression of GA [17]. Another phase three trial on ALK-001, a chemically modified vitamin A, is also being investigated for GA (NCT03845582). Furthermore, retinylamine, which lowers the concentration of all-trans retinal, has also been considered for AMD [18]. Another visual cycle modulator, Emixustat (NCT01802866), which inhibits RPE-specific protein RPE65 was shown to reduce the growth rate of small sized atrophy due to Stargardt disease but failed to show a benefit in GA [19]. The results of these studies show that further work is required to understand how anti-lipofuscin drugs will benefit eyes with AMD.

Further studies dissecting the qAF8 levels in ageing and various stages of AMD may provide new insight in AMD therapies. Using multimodal imaging, evaluating the differences of qAF8 levels in eyes with and without SDD may indirectly provide information on the relation of qAF8 and the visual cycle. In addition, the relation of qAF8 with the worsening health of RPE cells may be interrogated by its associations with optical coherent tomography (OCT) features, such as hyperreflective foci (HRF), incomplete and complete RPE and outer retinal atrophy (iRORA and cRORA, respectively) [20]. 

The aims of this study were to: (i) evaluate whether qAF8 levels vary with AMD progression; (ii) study whether qAF8 levels vary in eyes with and without SDD to investigate the relation of qAF8 with the visual cycle and (iii) interrogate whether OCT risk features of RPE cell death influence the qAF8 levels.

## 2. Materials and Methods

This single centre, prospective cohort study (PEONY study) was approved by the London-Chelsea Research Ethics Committee London REC 19/LO/0931. Written informed consent was obtained from all participants and the study followed the tenets of the Declaration of Helsinki. 

### 2.1. Setting

The National Institute of Health and Research Clinical Research Facility at a tertiary care hospital in London, UK.

### 2.2. Participants

Caucasians aged ≥50 years with varying severity of non-neovascular AMD in at least one eye and age-matched controls with healthy maculae were recruited for this study between August 2020 and July 2022. The inclusion criteria for non-neovascular AMD in at least one eye were a Snellen visual acuity of ≥6/18 with media clarity for qAF8 and other imaging. The exclusion criteria included any condition that, in the opinion of the investigator, could affect or alter visual acuity or retinal imaging, such as vitreous opacities, epiretinal membrane and other comorbid conditions such as diabetic retinopathy. 

### 2.3. Study Assessments

Age, gender and lens status were recorded. The multimodal imaging obtained for this study included near infrared reflectance (NIR), fundus autofluorescence (FAF), qAF8, en-face SD-OCT scans (97-line scans, 6 × 6 mm grid), enhanced depth imaging OCT (EDI-OCT), OCT angiography. These were conducted on Spectralis HRA + OCT, Heidelberg Engineering, Germany. 

### 2.4. Image Acquisition

Two trained operators acquired the images after mydriasis. A standard operating procedure was followed. In summary, with room lights were turned off and the NIR image was recorded first. In the qAF8 mode, the image was refocused to ensure a uniform signal over the entire field. During the set-up and focusing in the qAF8 mode, the fundus was exposed for 20 s to reduce visual pigment absorption. Each image was acquired in high-speed video mode, at least 2 images were recorded, each being 12 frames. All video images were examined for image quality and at least 9 of the 12 frames were selected. The selected frames were aligned and the mean image saved. The image quality was graded based on the criteria defined by von der Emde et al. (poor, acceptable or excellent) and all images with quality acceptable or above were included [21].

### 2.5. Image Analysis

The qAF8 images were analysed using the HEYEX software. The inter-operator agreement of qAF8 on five participants was measured first (kappa-0.89). The qAF8 levels were measured from the qAF8 segments, which refer to the middle ring of the Delori grid centered on the fovea that divides the area between 9° to 11° eccentricity from the fovea into eight segments. Vessels were automatically excluded from the analysis by the software. The threshold setting was manually adjusted if necessary. Phakic eyes were subsequently corrected for normative age-related optical media density [13,22]. No age-adjustment for ocular media absorption was applied for pseudophakic eyes [22]. 

The normalised grey values from the middle eight segments of the qAF8 image were averaged to provide the main outcome measure of this study (qAF8). 

### 2.6. Definition of Various AMD Categories

The qAF8 levels were compared across the various stages of AMD on colour photographs based on the Beckman classification [23]. The stages included: (i) no evidence of AMD or normal aging was defined as the presence of small drusen or druplets of <63 µm diameter; (ii) early AMD included eyes with a medium drusen, between 63 µm and 124 µm in diameter, with no pigmentary changes; (iii) intermediate AMD (iAMD) was defined as eyes with a drusen diameter of ≥125 µm or medium drusen with pigmentary changes; and (iv) eyes with GA [23]. This allowed us to investigate whether the qAF8 levels varied with a visible progressive loss of integrity of RPE on the colour photographs. 

Next, qAF8 was compared in the AMD categories classified by the multimodal imaging (MMI). Participants with a minimum of 5 SDD on different line scans on OCT confirmed on NIR were identified as a separate category, irrespective of the size of the drusen present. Eyes with SDD were further divided into stages: stage 1, defined as a diffuse deposition of granular hyperreflective material between the RPE-Bruch membrane band and the ellipsoid zone (EZ); stage 2, mounds of accumulated material sufficient to deflect inwardly the contour of the EZ; and stage 3, the presence of a conical appearance and breaking through the EZ [24]. AMD eyes without SDD were grouped based on their drusen size on the OCT, as reported by Kim et al. [25,26]. On the en-face OCT, a large drusen was defined as a drusen with a diameter ≥145 µm; the medium drusen diameters were between 100 µm and 144 µm; and the small drusen had diameters <100 µm on the OCT. Accordingly, the multimodal imaging classification of AMD included: (i) early AMD on the OCT, defined as maximum druse diameter <100 µm with no SDD or atrophy; (ii) intermediate AMD defined as having more than 1 druse with diameter 100–145 µm or at least 1 druse (>145 µm) without SDD or atrophy (cRORA), but may have HRF or iRORA; (iii) eyes with SDD; and (iv) eyes with atrophy (cRORA). We also investigated the relation of the drusen volume with the qAF8 values. The OCT scans were segmented by in-built automated Heidelberg software and manually corrected where necessary. 

Lastly, we compared the qAF8 values in the AMD eyes with varying degress of RPE cell loss identified on the OCT. These included comparing the qAF8 in eyes with: (i) no HRF, iRORA or cRORA; (ii) only HRF with no iRORA or cRORA; (iii) presence of iRORA with no cRORA; and (iv) cRORA. 

### 2.7. Statistical Analysis

The data were summarised using mean (SD) or median (IQR) for continuous variables and number (%) for categorical variables. The statistical differences in the log_e_ Mean qAF8 levels between the controls and individual AMD groups by multimodal imaging OCT and Beckman classification were compared after adjustment for the potential confounders of age, sex and lens status (pseudophakic vs. phakic). The stage of SDD, qAF8 and drusen volume were summarised based on the varying severity of RPE cell loss in AMD. The associations after adjusting for age, sex and lens status were quantified with odds ratio (95% CI) and *p*-value. Generalised Estimating Equations (GEE) with an exchangeable working correlation structure were used to account for the within-participant correlation among those with data from both eyes [27,28]. The qAF8 levels were log_e_-transformed to avoid estimating misleading associations from outliers. Model coefficient estimates were therefore interpreted in relation to the relative percentage increase rather than the absolute increase in the qAF8 values. Where log-transformations were applied to the independent variable qAF8 in the GEE models, the model estimates were interpreted in relation to the relative percentage increase in the independent variable. Spearmans correlation coefficient for clustered data, using the within-cluster resampling (WCR)-based method and standard errors based on fixed denominator, was used to test the correlation between age and mean qAF8 levels [29].

## 3. Results

### 3.1. Participant and Imaging Characteristics

The analysis sample included 105 eyes from 63 controls and 248 eyes from 176 AMD participants (Appendix A). These participants had a mean age of 70.3 years (SD 8.5), 138 were females (59.7%) and 60 (17.0%) eyes were pseudophakic (Table 1). 

### 3.2. Comparison of qAF8 Levels in AMD versus Control Eyes Aged ≥50 Years

The spearman correlation coefficient between age and qAF8 values in controls ≥50 years was ρ_WCR_ = −0.21; *p* = 0.08, in AMD eyes ≥50 years was ρ_WCR_ = −0.05; *p* = 0.49 (Appendix A) and in ≥65 years was ρ_WCR_ = −0.05; *p* = 0.54. Compared with the controls ≥50 years, the qAF8 values were significantly reduced in eyes with any AMD (241.6 [IQR 185.4–294.4] vs. 163.5 [IQR 122.6–216.6]; unadjusted difference= −25.2% [95% CI −33.0%, −16.5%]; *p* < 0.001; GEE model). The participant and imaging characteristics in AMD categories are presented in Appendix A. 

### 3.3. Comparison of qAF8 Levels in AMD Categories Based on Beckman Classification versus Control Eyes from Participants Aged ≥50 Years

There was a significant reduction in qAF8 in the AMD eyes versus control (−19.7% [95% CI −28.8%, −10.4%]; *p* < 0.001, Table 2). Statistical differences were found among all of the Beckman subgroups relative to the control eyes, in both the univariate analysis and following the adjustment for age, gender and lens status (adjusted % differences; Early AMD, −13.1% (−24.4%, −1%), *p* = 0.04; iAMD, −22.9% (−32.3%, −13.1%), *p* < 0.001; and GA −25.2% (−38.1%, −10.4%), *p* = 0.002). Significant differences were identified comparing early AMD with iAMD (% adjusted difference −11.4% [95% CI −0.10%, −21.3%]; *p* = 0.048, while the comparison between GA and early AMD failed to reach statistical significance in both the univariate and adjusted analysis (% adjusted difference −14.0% [95% CI −28.7%, 3.6%]; *p* = 0.11). No statistically significant differences were found between iAMD and GA.

### 3.4. Comparison of qAF8 Levels in AMD Categories Based on MMI versus Control Eyes from Participants Aged ≥50 Years

The qAF8 levels were found to be reduced in the eyes with AMD compared to the normal eyes, while the iAMD, SDD and GA groups are in close correspondence (Figure 1). In the adjusted analysis, differences were found between the OCT AMD subgroups relative to the controls, with the exception of early AMD (adjusted % differences; Early AMD, −5.8% (−18.9%, 8.3%); *p* = 0.40; iAMD, −26.7% (−36.2%, −15.6%); *p* < 0.001; SDD, −23.7% (−33.6%, −12.2%); *p* < 0.001; and cRORA, −26.7% (−39.3%, −11.3%), *p* = 0.001, Table 2). Significant differences were found when comparing early AMD to the other categories of AMD based on the MMI classification (iAMD; % adjusted difference of −21.9% [95% CI −32.6%, −9.5%]; *p* = 0.001), SDD; −19.0% [95% CI −30.4%, −5.7%]; *p* = 0.006), cRORA −22.0% [95% CI −36.1%, −4.8%]; *p* = 0.01). No significant differences were found when comparing the non-early categories of AMD. Figure 2 shows example qAF8 images of the participants with the same age across the different AMD groups.

### 3.5. Comparison of qAF8 Levels in AMD Eyes with and without Risk Factors of Progression to Geographic Atrophy or cRORA

Appendix A shows that an increase in the qAF8 levels was associated with a reduction in the odds of presenting with HRF, iRORA or cRORA (OR 0.91 [95% CI 0.85–0.97] per 10% increase in qAF8; *p* = 0.006). An increasing drusen volume in those without cRORA was associated with an increase in the odds of presenting with HRF, iRORA or cRORA (OR 2.00 [95% CI 1.47–2.73] per 0.1 unit increase in drusen volume; *p* < 0.001).

## 4. Discussion

Our study shows that the qAF8 levels are variable among the controls [2,30,31]. 

The mean qAF8 levels significantly decreased in AMD eyes (all AMD categories taken together) compared with the age-matched controls. The decrease in qAF8 may be explained by a change in the configuration of bisretinoids produced in AMD that may be less autofluorescent [32]. Alternatively, we can infer that the photodegradation of bisretinoids may be more rapid than the accumulation of photooxidised bisretinoids in AMD [32]. 

We also observed that RPE cell dysfunction appears early in AMD. Each AMD category was associated with reduced qAF8 compared to the age matched controls, with no significant differences in the qAF8 levels between them. These observations reflect early RPE dysfunction. Exocytosis may play a role [33]. As more RPE cell death occurs, more and more new RPE cells may undergo dysfunction, so the mean qAF8 may not be different between the AMD groups [16].

We also found that decreased qAF8 values are associated with eyes at risk of continuing RPE cell death [34,35]. Bird hypothesised that the reduced ability of RPE cells to degrade the outer segments may reduce lipid availability to the photoreceptor cells to produce outer segments and the shortening of outer segments would follow [36]. In turn, there may be reduced shedding and decreased phagosomal load in the RPE and reduced lipofuscin formation. 

The qAF8 levels did not vary significantly with the drusen volume in eyes with AMD [37]. Nonetheless, the drusen volume was found to be increased in people with HRF, iRORA and cRORA, in keeping with the fact that the RPE overlying the drusen tends to thin and disintegrate, leading to the appearance of HRF and consequent low qAF8 values [38]. 

Another hypothesis is that reduced qAF8 levels may suggest abnormal visual cycle as the spatial location of lipofuscin reflects those of the rods [2]. However, there was no statistical difference in the qAF8 levels between the AMD eyes with and without SDD. As SDDs are markers of a delayed rod intercept time, our findings do not suggest that rod dysfunction or changes in the visual cycle influences the qAF8 levels [39]. 

Although anti-lipofuscin interventions may be useful to reduce the toxicity of lipofuscin on the RPE cells, our study implies that very early intervention is required to prevent RPE dysfunction in people aged less than 70 years. After 70 years, there seems to be a normal decline in the RPE cell function based on the qAF8 values. Although late onset Stargardt disease can be confused with GA, there were no eyes with GA and high qAF8 in this study. 

Our study has several strengths. Firstly, our study is the largest study on qAF8 in AMD to date, and we included a large proportion of eyes with SDD and a wide age-range, particularly above 70 years. Both the Beckman and multimodal classifications were used to validate our findings. The participants with early AMD were grouped separately, rather than together with the healthy cohort, to understand the differences between them and healthy eyes. The multimodal classification allowed for the deep phenotyping of AMD. Thus, we could evaluate the changes in qAF8 in eyes with SDD and varying degrees of RPE loss. Our study is strengthened by a uniform and validated methodology of image acquisition and analysis and the use of robust statistics with adjustment for age and gender. 

However, our study has a few limitations. The intrinsic biologic limitations of qAF8 imaging cannot be ignored [16,40]. We had a large percentage of phakic eyes that were adjusted for the transmission factor, and this may have influenced the qAF8 levels. Additionally, we used only the middle qAF8 segments, which is not representative of the entire macula. Nonetheless, including other segments has its own limitations; therefore, the most validated measurement of qAF8 was used [41].

In conclusion, we show that there is a decline in qAF8 levels in eyes with AMD compared with age-matched controls. We have shown various lines of evidence that qAF8 is reduced during early AMD progression. The qAF8 levels seem related more to the loss of function and integrity of RPE cells rather than being due to abnormalities in the visual cycle. 

## Figures and Tables

**Figure 1 biomedicines-11-00560-f001:**
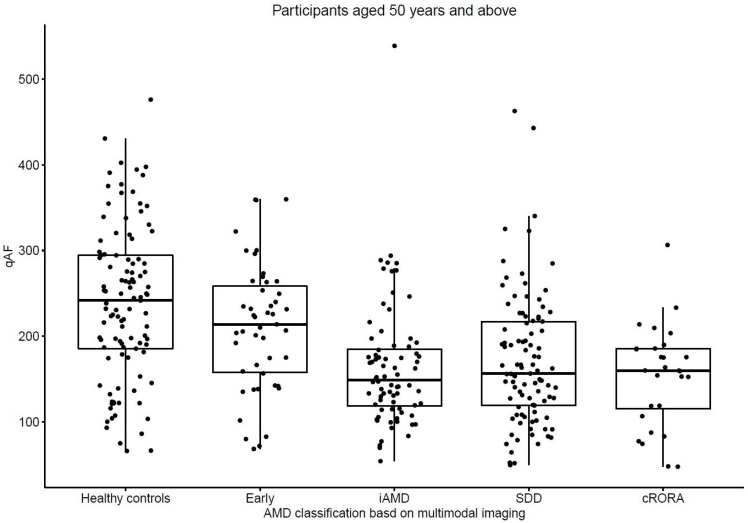
Mean qAF8 values in multimodal imaging categories of AMD versus age-matched controls. The multimodal imaging AMD classification shown on the *x*-axis. Comparison was with age-matched controls. Early-drusen diameter <100 µm and no SDD, intermediate—drusen diameter 100–144 µm or 1 drusen 145 µm or more and no SDD, SDD-eyes with SDD with no cRORA and final group-cRORA. Abbreviations: AMD—age related macular degeneration; cRORA—complete retinal pigment epithelium and outer retinal atrophy; iAMD—intermediate AMD; qAF8—quantitative autofluorescence; SDD—subretinal drusenoid deposits.

**Figure 2 biomedicines-11-00560-f002:**
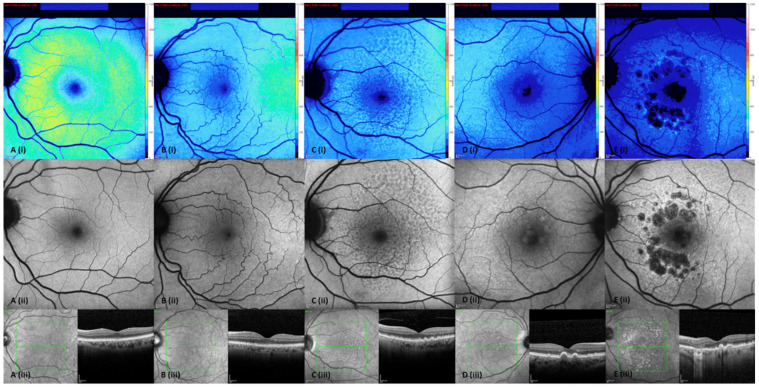
Multimodal imaging showing qAF8 changes across degeneration (AMD) categories in participants with same age. (**A**)—Controls; (**B**)—Early AMD; (**C**)—Subretinal drusenoid deposit; (**D**)—Intermediate AMD; (**E**)—Multifocal geographic atrophy. Images (i), (ii) and (iii) show the qAF8 image, fundus autofluorescence and IR + OCT images, respectively, for all groups. Abbreviations: AMD—age related macular degeneration; cRORA—complete retinal pigment epithelium and outer retinal atrophy; GA—Geographic atrophy; iAMD—intermediate AMD; IR- Infrared reflectance; OCT—Optical Coherence Tomography; qAF8—quantitative autofluorescence; SDD—subretinal drusenoid deposits.

**Table 1 biomedicines-11-00560-t001:** Participants and imaging characteristics in AMD categories and age-matched controls.

		AMD vs. Controls	
Variable	Overall, N = 353 Eyes of 231 Participants	Controls, N = 105 Eyes of 63 Participants	AMD, N = 248 Eyes of 176 Participants
Participant level ^a^			
Number of participants with bilateral eyes	122 (52.8%)	42 (66.7%)	72 (40.9%)
Age, years	70.3 (8.5)	64.2 (8.2)	72.4 (7.5)
Age, years			
<60	30 (13.0%)	20 (31.8%)	11 (6.3%)
60–69	65 (28.1%)	24 (38.1%)	44 (25.0%)
70–79	104 (45.0%)	15 (23.8%)	92 (52.3%)
≥80	32 (13.9%)	4 (6.3%)	29 (16.5%)
Gender			
M	93 (40.3%)	28 (44.4%)	68 (38.6%)
F	138 (59.7%)	35 (55.6%)	108 (61.4%)
Eye level ^b^			
Pseudophakic eyes	60 (17.0%)	7 (6.7%)	53 (21.4%)
Beckman AMD classification			
Normal	105 (29.7%)	105 (100.0%)	0 (0.0%)
Early	59 (16.7%)	0 (0.0%)	59 (23.8%)
iAMD	161 (45.6%)	0 (0.0%)	161 (64.9%)
GA	28 (7.9%)	0 (0.0%)	28 (11.3%)
AMD classification based on multimodal imaging			
Normal	105 (29.7%)	105 (100.0%)	0 (0.0%)
Early	47 (13.3%)	0 (0.0%)	47 (19.0%)
iAMD	76 (21.5%)	0 (0.0%)	76 (30.6%)
SDD	97 (27.5%)	0 (0.0%)	97 (39.1%)
cRORA	28 (7.9%)	0 (0.0%)	28 (11.3%)
Median qAF8 (IQR)	184.0 (132.2, 246.4)	241.6 (185.4, 294.4)	163.5 (122.6, 216.6)
Analysis in AMD eyes ^c^; n (%)			
Stage of SDD			
0	151/248 (60.9%)	0 (NA%)	151/248 (60.9%)
1	5/248 (2.0%)	0 (NA%)	5/248 (2.0%)
2	15/248 (6.0%)	0 (NA%)	15/248 (6.0%)
3	77/248 (31.0%)	0 (NA%)	77/248 (31.0%)
Indicators of RPE integrity			
Absence of HRF, iRORA or cRORA	153/248 (61.7%)	0 (NA%)	153/248 (61.7%)
Presence of HRF without evidence of iRORA or cRORA	16/248 (6.5%)	0 (NA%)	16/248 (6.5%)
Presence of iRORA without any evidence of cRORA	51/248 (20.6%)	0 (NA%)	51/248 (20.6%)
Presence of cRORA	28/248 (11.3%)	0 (NA%)	28/248 (11.3%)
Drusen volume in those without cRORA (N = 220 eyes)	0.44 (0.36, 0.54)	NA	0.44 (0.36, 0.54)

^a^ Participant level characteristics were based on total cohort of 231 participants. The number of participants across healthy and AMD do not add up to 231 (total number of participants in the cohort) due to bilateral eligibility in the cohort where eyes from the same participant may belong in both groups. ^b^ Eye level characteristics based on total analysed sample of 353 eyes. ^c^ Analysis in AMD eyes based on 248 eyes with AMD only, except drusen volume which has been analysed in 220 AMD eyes without cRORA. Abbreviations: AMD—age related macular degeneration; cRORA—complete retinal pigment epithelium and outer retinal atrophy; GA—Geographic atrophy; HRF—Hyperreflective foci; iAMD—intermediate AMD; iRORA—incomplete retinal pigment epithelium and outer retinal atrophy; IQR—Interquartile range; qAF8—quantitative autofluorescence; RPE—Retinal pigment epithelium; SDD—subretinal drusenoid deposits.

**Table 2 biomedicines-11-00560-t002:** Parameters associated with log-qAF8 mean adjusted for age, gender and lens status using GEE models.

Characteristics	N	Coefficient in Log-qAF8 Units	qAF8 Difference in %	*p*-Value
Controls vs. AMD	353			
Controls		-	-	
AMD		−0.22 (−0.34–−0.11)	−19.7% (−28.8%, −10.4%)	<0.001
Beckman classification	353			
Normal		-	-	
Early		−0.14 (−0.28–−0.01)	−13.1% (−24.4%, −1%)	0.04
iAMD		−0.26 (−0.39–−0.14)	−22.9% (−32.3%, −13.1%)	<0.001
GA		−0.29 (−0.48–−0.11)	−25.2% (−38.1%, −10.4%)	0.002
AMD classification based on multimodal imaging	353			
Normal		-		
Early without SDD		−0.06 (−0.21–0.08)	−5.8% (−18.9%, 8.3%)	0.40
iAMD without SDD		−0.31 (−0.45–−0.17)	−26.7% (−36.2%, −15.6%)	<0.001
SDD		−0.27 (−0.41–−0.13)	−23.7% (−33.6%, −12.2%)	<0.001
cRORA		−0.31 (−0.50–−0.12)	−26.7% (−39.3%, −11.3%)	0.001
Stage of SDD in those with SDD	97			
Stage 1 or 2		-		
Stage 3		−0.05 (−0.28−0.18)	−4.9% (−24.4%, 19.7%)	0.67
Stage of SDD	248			
0		-		
1		0.01 (−0.17–0.19)	1% (−15.6%, 20.9%)	0.88
2		0.05 (−0.22–0.33)	5.1% (−19.7%, 39.1%)	0.71
3		−0.05 (−0.17–0.07)	−4.9% (−15.6%, 7.3%)	0.38
Drusen volume in those without cRORA, per 0.1-unit increase	220	−0.01 (−0.05–0.02)	−1.4% (−4.6%, 1.9%)	0.40
Indicators of RPE integrity	248			
Absence of HRF without evidence of iRORA or cRORA		-		
Presence of HRF without evidence of iRORA or cRORA		−0.18 (−0.43–0.07)	−16.5% (−34.9%, 7.25%)	0.16
Presence of iRORA without any evidence of cRORA		−0.16 (−0.28–−0.03)	−14.8% (−24.4%, −3.0%)	0.01
Presence of cRORA		−0.13 (−0.30–0.05)	−12.2% (−25.9%, 5.1%)	0.15

Abbreviations: AMD—age related macular degeneration; cRORA—complete retinal pigment epithelium and outer retinal atrophy; GA—Geographic atrophy; GEE—Generalised estimating Equation; HRF—Hyperreflective foci; iAMD—intermediate AMD; iRORA—incomplete retinal pigment epithelium and outer retinal atrophy; qAF8—quantitative autofluorescence; RPE—Retinal pigment epithelium; SDD—subretinal drusenoid deposits.

## Data Availability

The data are not publicly available due to confidentiality issues and will only be available on reasonable request from the corresponding author.

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
