# Peer review of "Quantitative Autofluorescence in Non-Neovascular Age Related Macular Degeneration"

_biomedicines, 2023, doi:10.3390/biomedicines11020560_

Round 1
Reviewer 1 Report (Previous Reviewer 2)
The authors improved the manuscript and responded to questions raised by the reviewer.
Reviewer 2 Report (Previous Reviewer 3)
dear author:
Thanks for your revision for article.
The newly documents are good for the readers.
This manuscript is a resubmission of an earlier submission. The following is a list of the peer review reports and author responses from that submission.
Round 1
Reviewer 1 Report
Thanks for giving me a chance to review this manuscript. In this study, the authors assessed the changes in qAF8 levels in aging and in varying severity of non-neovascular AMD. This study can't be considered for publication due to some methodological problems. Here are some comments:
1. Results: Participant and imaging characteristics: "The analysis sample included 105 eyes from 63 controls and 176 eyes from 248 AMD participants (Figure S1)". However, in Table 1, they showed an Overall N = 353 eyes of 231 participants. 105 + 176 = 281, not 353 eyes.
Moreover, Table 1 shows AMD, N = 248 eyes of 176 participants. Previously they mentioned 176 eyes from 248 participants. It is confusing.
2. In the methods part: Participants subsection- "To study qAF8 in AMD versus age-matched controls, we only included people aged ≥50 years with normal fundus as controls".
However, in the result part: Age-related changes in qAF8 in control eyes aged 40 ≥years subsection. it was not in the inclusion criteria. how have they included participants less than 50? That means their inclusion criteria are wrong.
Moreover, I don't have any clue why they have shown only 120 participants' data in Figure 1. Why not all participant's data?
Reviewer 2 Report
This manuscript addresses a prospective cohort study to investigate changes in qAF8 levels in aging and in varying severity of non-neovascular AMD. The objectives of the study are well defined, and the introduction provides a state of art with appropriate references. Overall, I consider that despite the intrinsic biologic limitations of qAF8 imaging and the fact that the middle qAF8 segments are not representative of the entire macula, this manuscript contains enough data to evidence the conclusions.
Some questions
1) Why did we only include people aged ≥50 years with normal fundus as controls to study qAF8 in AMD versus age-matched controls, and not people aged ≥40 years with normal fundus as selected to determine age related changes of qAF8?
2) Exclusion criteria are somewhat subjective, as they depend on the opinion of the investigator. Couldn't you be more objective regarding these criteria?
3) In the results (Participant and imaging characteristics), please correct the folowing sentence:
176 248 eyes from 176 248 AMD participants (Figure S1).
Reviewer 3 Report
Dear authors:
Thanks for your working for this articles.
However, the methods (Page. 3,4,5) were not well written and the contests were difficult to read.
Besides, the results (Page 5.6.7.9 ) also made us confused.
Moreover, the scientific sound is low for the ophthalmologists and researchers .
Therefore, I suggested that the articles may be withdrawn.
Best regards.